# Chemical Recycling of Fully Recyclable Bio-Epoxy Matrices and Reuse Strategies: A Cradle-to-Cradle Approach

**DOI:** 10.3390/polym15132809

**Published:** 2023-06-25

**Authors:** Lorena Saitta, Giuliana Rizzo, Claudio Tosto, Gianluca Cicala, Ignazio Blanco, Eugenio Pergolizzi, Romeo Ciobanu, Giuseppe Recca

**Affiliations:** 1Department of Civil Engineering and Architecture, University of Catania, Via Santa Sofia 64, 95125 Catania, Italy; giuliana.rizzo@phd.unict.it (G.R.); claudio.tosto@unict.it (C.T.); gianluca.cicala@unict.it (G.C.); iblanco@unict.it (I.B.); eugenio.pergolizzi@unict.it (E.P.); 2INSTM-UDR CT, Viale Andrea Doria 6, 95125 Catania, Italy; 3Electrical Engineering Faculty, Gheorghe Asachi Technical University of Iasi, Dimitrie Mangeron Bd. 67, 700050 Iasi, Romania; rciobanu@yahoo.com; 4CNR-ICPB, Via Paolo Gaifami 17, 95125 Catania, Italy; giuseppe.recca@cnr.it

**Keywords:** circular economy, chemical recycling, fully recyclable epoxy resin, thermoplastic/epoxy resin blend, bio-based resin, recycled matrices reuse

## Abstract

Currently, the epoxy resin market is expressing concerns about epoxy resins’ non-recyclability, which can hinder their widespread use. Moreover, epoxy monomers are synthesized via petroleum-based raw materials, which also limits their use. So, it is crucial to find more environmentally friendly alternative solution for their formulation. Within this context, the aim of this paper is to exploit a Cradle-to-Cradle approach, which consists of remodeling and reshaping the productive cycle of consumer products to make sure that they can be infinitely reused rather than just being recycled with a downgrading of their properties or uses, according to the principle of the complete circular economy. Indeed, after starting with a fully-recyclable bio-based epoxy formulation and assessing its recyclability as having a process yield of 99%, we obtained a recycled polymer that could be reused, mixing with the same bio-based epoxy formulation with percentages varying from 15 wt% to 27 wt%. The formulation obtained was thoroughly characterized by a dynamic-mechanical analysis, differential scanning calorimetry, and flexural tests. This approach had two advantages: (1) it represented a sustainable disposal route for the epoxy resin, with nearly all the epoxy resin recovered, and (2) the obtained recycled polymer could be used as a green component of the primary bio-based epoxy matrix. In the end, by using replicated general factorial designs (as statistical tools) combined with a proper optimization process, after carrying out a complete thermo-mechanical characterization of the developed epoxy formulations, the right percentage of recycled polymer content was selected with the aim of identifying the most performing epoxy matrix formulation in terms of its thermo-mechanical properties.

## 1. Introduction

Epoxy resin is among the most used and commercialized thermosets due to its outstanding properties, such as its easy and cost-effective processing, chemical resistance, high durability, insulation, adhesiveness, low thermal expansion, and good thermal and mechanical properties [1,2]. These properties depend on epoxy resin’s constituents. For example, its chemical resistance is due to the presence of ether linkages, its adhesive property is attributable to the presence of epoxy and hydroxyl groups, and its rigidity is given by the presence of bisphenol A [3].

Due to their interesting properties, epoxy resins have different applications in several industrial fields; they are used in paints and coatings, adhesives and sealants, matrices for fiber-reinforced composites, electrical and electronic components, marine sector, wind turbines, and so on. The epoxy market was estimated at USD 22.9 billion in 2021, and it is expected to expand at a compound annual growth rate (CAGR) of 7.3% from 2022 to 2030 [4]. Even though epoxy resins have become predominant in many application areas, concerns are growing about their widespread use because of their non-recyclability once they have reached their end of life (EoL) [5]. This peculiarity can be traced back to their permanent covalent intermolecular cross-link bonds. For this reason, unlike thermoplastic materials, once epoxy resins are permanently cross-linked, they cannot be fused, solubilized, or re-processed [6]. Hence, at their EoL, only two mainly options are available for their disposal, i.e., incineration and/or landfills [7]. However, these two alternatives are becoming more and more restricted and banned, since they are considered to have a serious negative impact on the environment. This is the reason why both industry and academia are looking for more environmentally friendly recycling routes to reduce the environmental impact of epoxy resins [8]. Several researchers have already presented state-of-the-art papers which propose different recycling strategies for epoxy composites [9,10,11]. Currently, the main ones that have been proposed are chemical, mechanical, and thermal treatments. Chemical processes are frequently used to recover fibers after the depolymerization of the matrix into monomers or petrochemical feedstock. Mechanical recycling is mainly based on the crushing, grinding, or milling of epoxy composites to reduce them into smaller pieces, which can then be further treated and used as fillers. Lastly, thermal recycling implies the combustion of the resin matrix using a high temperature, thereby promoting the recovery of the fibers. However, all these strategies have a common drawback, which is the absence of the recovery of the epoxy matrix. To overcome this issue, the company Connora^®^Technologies in 2012 filed a patent (US Patent 2013/0245204 A1) [12] that addressed the issue of epoxy resins’ EoL treatment by programming their recyclability before they are cured. In other words, this company used an approach that is now called a ‘design for recycling’. This goal was achieved by curing the epoxy matrices using novel acetal-based molecules with built-in acid-cleavable acetal bonds that become part of the epoxy-cured network. After curing, these molecules drive the selective cleavage of the thermoset’s network, transforming it into a reusable oligomer when the epoxy is treated under mild conditions [13]. The first recycling process that was proposed by Connora^®^Technologies has recently been optimized, becoming more environmentally friendly [14]. For the first time, the authors thoroughly characterized the chemical structure of the recycled thermoplastic material obtained from selective cleaving. Knowledge of this chemical structure is pivotal to developing reuse strategies. A feasible approach was proposed in this paper that was based on the chemical reaction of recycled material with an isocyanate to obtain polyurethane.

In this work, we propose an alternative reuse strategy that starts from the produced recycled polymer (rTP), encompassing a truly circular economy (CE) approach and relying on the cradle-to-cradle (C2C) concept. The latter approach is gaining ground in the polymer field with the aim of developing innovative recycling technologies that can help establish a circular economy and sustainable, benign environment [15]. The C2C philosophy was first developed by McDonough and Braungart [16], and this approach aims to remodel and reshape the productive cycle of consumer products to make sure that they can be infinitely reused rather than just being recycled with a downgrading of their properties or uses [17]. Indeed, recycling routes for products that reach their EoL are seen as downgrading, because valuable materials’ properties are reduced with each cycle [16]. So, once a product is introduced in the production cycle, clear planning is necessary to clearly define what to do with the product itself after its use to avoid these reductive cycles [18,19]. In line with the CE/C2C methodology, in this experimental work, the reuse of recycled polymers was investigated. An rTP was added to uncured epoxy resins with contents varying from 15 wt% to 27 wt%, according to the well-known ‘thermoplastic/epoxy blending’ approach [20,21,22]. These blends have been widely investigated since the early 1980s to overcome some of the limitations of epoxy resins. To modify epoxies, many different types of additives have been used over the years, such as rubbers, inorganic glasses, polyurethanes, acrylates, and ductile engineering thermoplastics, mainly used as toughening agents [23,24,25,26,27]. Thermoplastic blending with epoxies can also be used to obtain low-shrinkage resins [28] and to control resin porosity [29]. Adding rTPs to epoxy allows us to reuse the recycled product as a higher-value product, while also reducing the consumption of virgin epoxy monomers.

From this point of view, many studies already present in the literature have proposed the synthesis of bio-based epoxy monomers from vegetable oils, lignin, eugenol, and natural acids [30,31,32,33,34]. However, to the best of our knowledge, none have yet attempted to reuse a recycled product derived from the recycling process of an epoxy system back into the epoxy system itself with the aim of reducing the use of petroleum-based raw materials. In our experimental work, the negative environmental impact was further reduced by selecting a bio-based epoxy prepolymer characterized by a biocarbon content equal to 28% obtained from pine oils.

In the present paper, bio-based epoxy prepolymers and different formulations of rTPs with contents varying from 15 wt% to 27 wt% were processed with different cure cycles. All the systems were thoroughly characterized by thermo-mechanical testing. All the collected data by the test runs were statistically analyzed via a replicated general factorial design. In the end, the best formulation was identified using an optimization process relying on the desirability function approach.

## 2. Materials and Methods

### 2.1. Materials

The epoxy resin used for this study is a bicomponent system purchased from R*Concept (Barcelona, Spain). Part A is a bio-based epoxy prepolymer (EEW = 180–190 g/eq), named Polar Bear, characterized by a biocarbon content of 28% deriving from pine oils (expressed as a fraction of the total organic carbon content). The latter was determined according to the ISO/IEC 17025:2017 PJLA #59423 standard. Part B is an amine hardener named RecyclamineTM R*101 (AHEW = 52.5 g/eq). A cleavable acetal group is embedded in the epoxy resin system’s chemical structure, which can be selectively cleaved under mild acidic conditions, thus achieving the epoxy resin’s network breakage and, in turn, its recyclability.

Both part A and part B of the formulation were liquid at room temperature, hence making the mechanical mixing process easy.

For the chemical recycling process, pure acetic acid from VWR International S.r.l., Milan, Italy and ammonium hydroxide (28.0–30.0% NH_3_ basis) from Sigma-Aldrich (Merk Life Science S.r.l., Milan, Italy) were used.

### 2.2. Virgin Epoxy Resin Formulation

In accordance with a previous study [14], the virgin epoxy resin formulation was prepared by mixing part A and part B, with the content of the latter equal to 22 phr (per hundred resin). The formulation was then degassed to remove air bubbles using a planetary mixer (Thinky mixer ARV310, THINKY U.S.A., LH, CA) set at 3000 rpm for 5 min with a constant vacuum of 0.2 kPa. The degassed resin was then poured into a silicon mold. The formulation was left to cure at room temperature (i.e., 25 °C) for 24 h, and then it was post-cured at 100 °C for 3 h. The cure cycle was selected according to the results reported by authors previously [14]. The cured samples were obtained as bar specimens (80 mm × 10 mm × 4 mm).

### 2.3. Chemical Recycling Process

The cured specimens were chemically recycled using an optimized chemical recycling procedure as proposed elsewhere [14,35]. Briefly, 10 g of the cured epoxy matrix was immersed into a 300 mL acetic acid aqueous bath (75% v/v) at 80 °C for 90 min for its dissolution. Afterwards, the final solution was roto-evaporated at 60 mbar and 60 °C to recover the acetic acid and obtain a final volume of 75 mL of a more concentrated solution. This was then neutralized with 150 mL of ammonium hydroxide solution, obtained by mixing distilled water and ammonium hydroxide (28–30% NH_3_ basis) in a 1:1 ratio. During this phase, the rTP started to precipitate, and it was recovered, washed with distilled water, and dried at 50 °C for 24 h under vacuum conditions. Finally, the rTP was pulverized by using a mortar and pestle.

The yield for the chemical recycling process was equal to 99%, and it was evaluated using the formula reported below:(1)$$yield\left(\%\right)=\frac{W_{f}^{\left(rTP\right)}}{W_{i}^{\left(E\right)}}\times 100$$
where $W_{f}^{\left(rTP\right)}$ is the final weight of the recycled thermoplastic (once dried), and $W_{i}^{\left(E\right)}$ is the initial weight of the initial epoxy resin.

According to previous results obtained by the authors of [14], the obtained recycled polymer showed a glass transition temperature of 76 °C and molecular weight equal to 15,000 Da.

### 2.4. Epoxy Resin Formulation by Reusing the Recycled Polymer

Recycled thermoplastic (rTP) has been extensively characterized in our previous work [14]. Its chemical backbone (see Figure 1c) presents a similar structure to that of the epoxy prepolymer (Figure 1a) with diphenyl moieties and pendant hydroxyl groups per each repetitive unit. The reuse of the self-standing thermoplastic matrix has been previously demonstrated [14,36,37,38]. Therefore, to demonstrate an alternative reuse strategy for the recycled product, different formulations combining the epoxy prepolymer (i.e., Polar Bear) with the rTP were prepared. To ensure the recyclability of the synthetized thermosets, the rTP was first mixed with the cleavable ammine (RecyclamineTM R*101) (Figure 1b) and then used as a mixture. This pre-mixing procedure allowed us to obtain a liquid mixture at room temperature, while rTP was obtained as a solid after recycling.

Due to their low toxicity and availability, carboxylic acids [39], alcohols [40], and anhydrides [30,41,42] are widely applied as hardeners for epoxy manufacturing [39,43,44,45]. Therefore, recycled thermoplastics (rTPs) can be used, because the pendant -OH groups, as a co-curing agent, partially replace the cleavable amines. This approach allows us to obtain greener and more sustainable thermosets, as the standard Recyclamine is petroleum-based. The hydroxylic groups in the repetitive unit of the rTP oligomers can co-react during the curing process with the epoxy monomers mainly by two mechanisms: (a) ring opening of the glycidyl ethers by hydroxylic groups; or (b) speeding up of the reaction between epoxy and amine. The latter has been widely recognized and described in detail in a few works [46,47]. Bowen et al. [47] described, for example, how the gel time decreases between epoxies and amines with the increase in added hydroxylic groups per alcohol unit to the formulation. The effect of the hydroxylic groups on the reactivity between epoxies and amines is strongly related to the formation of highly unstable epoxy sites, due to the hydrogen bonding between the oxirane and hydroxyl groups. Figure 2a shows the generally accepted mechanism for -OH reaction in the epoxy–amine curing process. Additionally, the reactions between the reactive sites of Polar Bear and the curing blend (rTP + RecyclamineTM R*101) are represented in Figure 2b. As can be observed, the formation of ether groups is related to the cross-link point obtained by the reaction between the rTP and the epoxydic groups. Therefore, the rTP used in this work has two roles: it behaves as a curing agent, and it acts as a thermoplastic modifier in the blend. Similar behavior has been previously reported for reactive polyethersulphone copolymers [45].

To investigate the influence of the rTP on the thermo-mechanical properties of the epoxy blends, three formulations were prepared by curing the epoxy monomer named Polar Bear with a mixture of rTP and RecyclamineTM R*101. The molar ratio between the epoxy prepolymer and the hardening mixture (RecyclamineTM R*101 + rTP) was kept fixed to 1:0.46. Moreover, the percentage of rTP varied from 15%wt up to 27%wt. For all the blends, the following procedure was used: (i.) the selected amount of rTP was mixed with Recyclamine at 70 °C to ensure full dissolution; (ii.) the epoxy prepolymer was then added and mixed under constant degassing in a vacuum using a centrifugal mixer (Thinky mixer ARV310, THINKY USA, Laguna Hills, CA); and (iii.) the obtained formulation was poured into a silicone mould. Once the epoxy/thermoplastic blends were ready, they looked homogeneous, with the rTP totally dissolved in the epoxy resin.

Two cure cycles were analyzed: curing at 25 °C for 24 h (C1) and curing at 25 °C for 24 h followed by an additional post-curing step at 100 °C for 3 h (C2). The samples were tested after each curing cycle to evaluate the effect on the resin properties.

### 2.5. Thermo-Mechanical Characterization

The cured samples, obtained by mixing different amounts of rTPs, were characterized in terms of their thermo-mechanical properties to evaluate the range of applications for their reuse from the cradle-to-cradle perspective.

The rTP content added (named factor A) and the curing cycle (named factor B) were selected as independent variables for the study, and their combined effect was investigated using a replicated general factorial design. In detail, factor A was varied among three different levels (a = 3) of {15, 21, 27} %wt. Conversely, factor B was varied among two different levels (b = 2) of {C1, C2}. Four different responses (dependent variables) were investigated: (i.) glass transition temperature (T*_g_*), (ii.) flexural strength, (iii.) flexural modulus, and (iv.) deformation at break. The response parameters were determined using dynamic mechanical analysis (DMA), differential scanning calorimetry (DSC), and flexural test (ASTM D790), which are described below. For each experimental study, the number of replications was set equal to n = 5. Therefore, N = a · b · n = 30 runs were carried out for each considered experimental plan. In the end, once the response parameters were collected, an analysis of variance (ANOVA) was performed to investigate the statistical significance of each considered factor and their interaction. The four different experimental plans performed in the study are reported in Table 1.

Finally, to identify the best-performing formulation in terms of the investigated responses, an optimization was carried out by using the approach of desirability functions. This general approach first converts each response $y_{i}\;$into an individual desirability function $d_{i}$ that varies over the following range:$$0\leq d_{i}\leq 1$$
where if the response is at its goal or target, then $d_{i}=1$, while if the response is outside of an acceptable region, $d_{i}=0$. Then the design variables are chosen to maximize the overall desirability [48]. To carry out this analysis, precise constraints were fixed in terms of the maximization of T*_g_*, flexural strength, flexural modulus, and deformation at break. Factors A (rTP content) and B (curing cycle) were varied within the considered experimental range, i.e., among all the considered levels. The set constraints for the optimization process performed are summarized in Table 2.

#### 2.5.1. Thermal Analyses: Dynamic Mechanical Analysis (DMA) and Differential Scanning Calorimetry (DSC)

A DMA was used to measure the T*_g_* for each formulation with different rTP contents and different curing cycles. These analyses were run using a TRITEC 2000 dynamic mechanical thermal analyzer (Triton Technology, Leicestershire, UK) on samples with size of (10 × 6 × 4) mm^3^. Each analysis was performed in single cantilever deformation mode, since this is a well-known mode for characterization through glass transitions [49]. Each analysis was run in the temperature range between 25 °C and 150 °C with a heating rate equal to 2 °C/min. The displacement was set at 200 μm, and the frequency was 1 Hz. Five samples were tested for each investigated scenario. Next, the average and the standard deviation were evaluated for the T*_g_* using the following approach: tanδ versus temperature curves were plotted for each investigated formulation, and the T*_g_* was measured as the tanδ peak temperature.

DSC analysis was carried out to confirm the obtained results from DMA analysis and to understand the effect on the cross-linking process of the addition of rTP. The calorimetric measurements were run using a Shimadzu DSC-60 (Shimadzu, Kyoto, Japan) instrument. For each scan, about 6 mg of sample was inserted into 40 μL sealed aluminum crucibles. Each sample was heated from 25 °C up to 250 °C at a rate of 20 °C/min in air.

#### 2.5.2. Mechanical Characterization: Flexural Test

The flexural properties of each thermoset formulation containing varying contents of rTP were measured according to the ASTM D790 standard. The characterized samples had a size of (80 × 10 × 4) mm^3^. They were mechanically tested using an Instron 5985 universal testing machine (Instron, Milan, Italy) equipped with a load cell of 10 kN, in strain control mode, with a span length of 60 mm and a speed of 2.0 mm min^−1^. Five samples were tested for each investigated scenario measuring the flexural strength, the flexural modulus, and the deformation at break; next, the average and the standard deviation were evaluated for each of them.

### 2.6. Chemical Characterization

#### Fourier Transform Infrared Spectroscopy

The evolution of the chemical groups by increasing the rTP amount within the formulation was examined via Fourier transform infrared spectroscopy. The spectra, studied with a Perkin Elmer Spectrum 100 UATR (Waltham, MA, USA) in attenuated total reflectance (ATR) mode, were recorded in the range between 4000 and 650 cm^−1^ with 32 scans and a resolution of 4 cm^−1^.

### 2.7. Morphological Characterization: Scanning Electron Microscopy

SEM (scanning electron microscopy) micrographs were obtained using an SEM EVO 15 (Zeiss, Cambridge, UK). The post-cured samples were fractured in liquid nitrogen, and then the cryofractured surfaces were etched with stabilized Tetrahydrofuran ACS reagent (VWR International S.r.l., Milan, Italy) for 20 min before performing the sputter-coating process. The etching process was exploited to etch the rTP phase, thus increasing the contrast between the two phases (i.e., epoxy matrix and rTP) when phase separation occurred. The specimens were gold-sputtered with an Agar Sputter Coater AGB7340 sputter coater machine (Assing Italy) after the etching treatment. The SEM analysis was run at different magnifications: 100×, 300×, and 500×. The electron source used was an LaB6 (Lanthanum Hexaboride) emitter, while the electron high tension (EHT) value was set at 25 kV.

## 3. Results and Discussion

### 3.1. Thermal Characterization

The obtained results from the DMA test for the T*_g_* evaluation of each investigated scenario are reported in Table 3, while the tanδ versus temperature plots are shown in Figure 3.

The combined effect of the T*_g_* of the rTP content added (factor A) and the curing condition used (factor B) was rationalized using an ANOVA analysis, whose results are summarized in Table 4, Table 5, Figure 4 and Figure 5. According to the ANOVA analysis, A and B are influential factors (*p*-value* < 0.0001) for the T*_g_* value, as can be inferred from the effects diagram for the considered parameter (Figure 4). Furthermore, it also involved a significant AB interaction (*p*-value* < 0.0001). No anomaly was detected in the model adequacy check on the residuals, as shown in Figure 4. Finally, due to the very high value for the R^2^ parameter, equal to 0.9980 (Table 5), most of the variability observed for the collected *T_g_* values is attributable to the influential factors’ variation.

According to the effects diagram for the T*_g_* (Figure 4), it is possible to assume that the additional post-curing step at 100 °C for 3 h allowed us to obtain the highest value for the T*_g_* overall, thus confirming a higher cross-linking density for the epoxy resin system with an rTP content ranging between 15 and 27%wt. The identified effect of the post-curing process, which allowed us to achieve higher T*_g_* values, is consistent with the behavior of the cross-linking density previously found for the virgin epoxy matrix [14,35]. On the other hand, by adding increasing the rTP percentage content, the T*_g_* sharply decreased by about 18% at first, i.e., from 0%wt to 15%wt. Meanwhile, by increasing the rTP content, the T*_g_* continued to decrease, but in a less marked way (about 3%). This trend has also been reported in the literature when adding plasticizer within epoxy systems [50,51]. The decrease in the T*_g_* by increasing the rTP content in the thermoset formulation can be explained by the lower reactivity of the recycled thermoplastic compared to that of the commercial recyclable amine and by the dilution effect observed for thermoplastic/thermoset blends [52,53]. Both of these can lead to a lower cross-linking density of the final epoxy network. Indeed, due to its linear chemical structure, the recycled thermoplastic introduced within the epoxy network can increase intermolecular mobility, involving a decrease in the T*_g_*. In previous findings, the main molecular effect related to the interaction of a plasticizer with an epoxy network has been shown to be a broadening of its viscoelastic response [54]. However, in our study, no spread for the loss tangent (tanδ) peak over a wide range of polymers was identified (see Figure 3) by adding an increasing quantity of rTP to the epoxy formulation. Thus, it can be assumed that there is not a great amount of heterogeneity in the thermoset structure in the studied rTP’s modified epoxies. This is due to two causes: first, no epoxy and amine groups remain unreacted in the studied systems, contrary to what has been observed in the literature for the addition of plasticizer molecules in thermosets [54]; and second, the epoxy network formed in the rTP’s modified epoxy is homogenous because the chemical structure of the rTP is quite similar to that of the unmodified epoxy, both being derived from the same source. These findings also suggest that the rTP added to the virgin epoxy resin co-reacts with the epoxy/amine system, avoiding any significant reactivity reduction as can be observed with the addition of unreactive thermoplastics [55]. This finding is reasonable since the rTP is obtained from the initial epoxy network through a chemical cleavage process, leading to the formation of an oligomer with reacting -OH groups. In order to confirm such findings, a DSC analysis was carried out for each considered scenario. The obtained results are summarized in Figure 6.

The exothermic peak of the DSC thermogram, which represents the area corresponding to the polymerization process, showed only a slightly lower variation in the peak temperatures for the increasing rTP content (Table 5). Therefore, the addition of the rTP only slightly affected the reaction temperature (Table 5). Figure 6 summarizes the DSC analysis for the cured samples after the C1 and C2 cycles. After the C1 cure cycle, only a slight residual heat can be observed from the DSC. Fully cross-linked epoxy matrices, with no residual exotherm heat release, were found after the C2 cycle. This finding supports the conclusion that adding rTP has a minor effect on the overall resin reactivity.

### 3.2. Mechanical Characterization

The representative flexural stress versus flexural strain curves obtained for each investigated epoxy resin formulation with a variable content of rTPs, ranging between 15%wt and 27%wt, and exploiting two different cross-linking curing cycles (C1 and C2), are shown in Figure 7.

The data gained from the mechanical characterization, i.e., the flexural properties found for all the investigated scenarios, are summarized in Table 6. Furthermore, the results obtained for the flexural strength, the flexural modulus, and the deformation at break are reported in the bar plot charts in Figure 8, Figure 9 and Figure 10, respectively.

The ANOVA analysis was carried out to unveil the combined effect of factor A and factor B on the collected data for the flexural properties. According to the ANOVA study results for the flexural strength, which are reported in Table 7, both factors A and B are influential for the investigated response (*p*-value* < 0.0001). Moreover, even their interaction AB was influential (*p*-value* < 0.0001). According to the normal probability plot reported in Figure 11, no anomalies were found in the model adequacy check on the residuals. Since the R^2^ parameter had a high value, 0.9746 (Table 7), most of the variability observed for the flexural strength data is due to the variation in the influential factors found with the ANOVA analysis that was carried out.

According to the effects diagram (Figure 12), it is possible to assume that the post-curing process enhanced the flexural properties for both the formulations containing rTP in the range between 15 and 21%wt. Indeed, the additional post-curing process for the latter two levels of factor A allowed us to increase the considered property about 45% and 80% for the 15%wt and 21%wt rTPs, respectively. Compared with the results obtained for the virgin epoxy resin [35], the addition of the 15%wt and 21%wt rTPs allowed us to increase the flexural strength about 53% and 47%, respectively. This is similar to the findings of J. Wu et al. [56], and embedding the linear structure of the rTP is more flexible as the result of the thermoset network. However, it must also be highlighted that the improvement rate for the flexural strength decreased for an rTP content of 27%wt. In particular, for the parameter combination {A: 27%wt; B: C2}, the investigated response decreased by about 55% when compared to that of the virgin epoxy system cross-linked with the same curing cycle. The same system {A: 27%wt; B: C2} showed an even higher drop of about 70% when compared to that of the epoxy formulation with 15%wt and 21%wt rTP contents. This finding is explained as a combination of the higher plasticization of the epoxy matrix as well as the result of the phase separation, as demonstrated by the SEM analysis reported in the next section.

Following the same approach described for the first two responses (i.e., the T*_g_* and flexural strength), the combined effect of factor A and factor B on the flexural modulus can be determined. The ANOVA study results are reported in Table 8. From the latter, it can be deduced that, similarly to previous findings, factor A, factor B, and their interaction AB are influential for the considered response (*p*-value* < 0.0001). Furthermore, as R^2^ = 0.9270, it can be determined that most of the variability found for the collected data for the flexural modulus parameter is related to the variation in the influential factors found inthe ANOVA. Figure 13 shows the normal probability plot, from which it can observed that no anomalies were uncovered from the model adequacy check on the residuals.

In Figure 14, a diagram of the effects related to the flexural modulus is shown. It can be inferred that an increasing amount of rTP content caused a reduction in the flexural modulus if only the first C1 curing cycle was applied. This behavior can be explained as the result of the formation of a partially cross-linked epoxy network (which had already been proven by the DSC analysis) upon the first curing cycle C1 as well as the presence of the linear rTP chains within the epoxy network, which lead to a lower stiffness of the matrix. On the other hand, by comparing the flexural modulus of the unmodified epoxy matrix cured at 25 °C for 24 h [35] with the thermoplastic-modified thermoset, an increase in the modulus was observed. In detail, in the C1 curing cycle, it was equal to 1.35 GPa for the initial epoxy system. So, increases of 155%, 121%, and 86% were found by adding rTP contents of 15, 21, and 27%wt, respectively.

Applying the second cure cycle (C2) and focusing on the investigated scenario {A: 15%wt; B: C2}, the prevalent influence of factor A can be highlighted, since the additional post-curing step at 100 °C for 3 h allowed us to achieve an increase of about 1.7% in the flexural modulus. Compared to the results obtained for the virgin thermoset [35], in which the post-curing step involved an increment of about 30% for the considered parameter, the latter increment was slightly attenuated by the addition of 15%wt rTP. However, when considering the same curing cycle (C2), the addition of the 15%wt rTP in the epoxy formulation guaranteed an increase of 100% for the mechanical property considered, when compared to the virgin formulation (i.e., with no rTP addition). Considering the second scenario {A: 21%wt; B: C2}, a sharp increase of 49% for the investigated parameter was detected when compared to that of the C1 cross-linking procedure only. Moreover, considering the same curing conditions (C2), this formulation allowed us to enhance the flexural modulus about 154% when compared to that of the virgin thermoset. Regarding the third scenario {A: 27%wt; B: C2}, the flexural modulus was 9% higher when compared to that of the C1 cross-linking procedure. In this case, the increase obtained for the flexural modulus by switching from the C1 to C2 curing cycle was mitigated by the high content of rTP added to the thermoset formulation. This behavior confirmed the results found from the DMA tests carried out, i.e., a high amount of the recycled polymer in the formulation itself caused an excessive plasticization effect on the thermoset network.

For the results discussed so far, a common feature can be highlighted: the addition of rTP to the virgin thermoset allowed us to improve the flexural modulus, even though this varied depending on the rTP quantity added. A similar trend was found by M.I. Giannotti et al. [57], who performed a series of flexural tests on thermoplastic-modified epoxy resin. In detail, they mixed a thermoset material made of a diglycidyl ether of bisphenol-A (DGEBA) epoxy monomer and an aromatic amine curing agent 4,4′-di-aminodiphenyl sulphone with polysulphone (PSU) as the thermoplastic material. They found there was an increase of about 4% in the flexural modulus by moving from the unmodified to the modified epoxy resin. Furthermore, analogous results were achieved by J. Essmeister et al. [58], who found that as increasing quantity of poly(p-phenylene pyromellitimide) thermoplastic filler was added into a commercial epoxy system consisting of a DGEBA prepolymer (diglycidylether of bisphenol-A) and an amine-hardener formulation, the flexural modulus increased. However, in our study, a threshold value for the rTP content was found, because increasing its value above 21%wt caused the start of a decrease in the flexural modulus value, which nevertheless remained higher than the one collected for the virgin thermoset matrix.

Finally, the ANOVA results found for the fourth dependent variable, the elongation at break, are reported in Table 9. In agreement with the obtained results, both factors A and B are not influential for the elongation at break (*p*-value∗ > 0.0001). Conversely, their interaction AB was influential (*p*-value∗ < 0.0001). Next, considering the high value found for the R^2^ parameter (R^2^ = 0.9566) (Table 9), most of the variability observed for the acquired data (the elongation at break) is due to the variation in the influential factor (AB).

Lastly, from the normal probability plot related to the response elongation at break shown in Figure 15, no anomalies were found in the model adequacy check on the residuals.

The diagram of effects related to the elongation at break is reported in Figure 16. Similar to the findings achieved for the flexural strength, the C2 curing cycle led to an increase in the elongation at break for the first two levels considered for parameter B, for which increases for the investigated parameter of 54% and 49% were found for rTP contents added to the epoxy matrix equal to 15%wt and 21%wt, respectively. Conversely, in correspondence to the scenario {A: 27%wt; B: C2}, since the rTP content exceeded the previously identified threshold value, the considered response was drastically reduced. So, confirming the findings already discussed for the flexural strength, an excessive rTP content can lead to an agglomeration effect of the thermoplastic matrix on the thermoset resin, due to the lack of homogeneity involving an early feature. The same explanation can also justify the fact that the elongation at break is cut in half by moving from the unmodified epoxy matrix to the thermoplastic-modified thermoset matrix at different contents of rTP. A similar behavior was detected by J. Wu et al. [1], who investigated the flexural properties of a thermoset matrix by adding various contents of polyetheretherketone (PEEK) thermoplastic fillers.

To validate the above-described reaction mechanism, the chemical structures of the cured formulations were analyzed by infrared spectroscopy, and the related spectra are reported in Figure 17. The three different formulations were thus compared with the reference (Polar and R*101 without any addition of rTPs), focusing on the following ranges: 3800–2500 cm^−1^, 1800–1400 cm^−1^, and 1200–900 cm^−1^. By increasing the amount of the thermoplastic, an increase in the intensity of the band at 3396 cm^−1^ [59] was observed. The latter is associated to the stretching of the O–H group characterizing the rTP. The higher the percentage of rTP, the wider and more intense the band is. Another interpretation is that it is associated with the strong hydrogen bonding between the unreacted hydroxylic group and both the nitrogen and oxygen atoms present in the polymeric network. The peaks detected at 2915 cm^−1^ and 1506 cm^−1^ are linked to the stretching and bending of the C–H groups in aliphatic ethers, respectively [60]. From the mechanisms reported in Figure 2, the formation of ethers is mainly associated to the ring opening of epoxies by the —OH group of the rTP. The intensity of both increases by increasing the rTP amount within the formulation; thus, it is related to the more extended curing reaction between Polar Bear and the recycled thermoplastic used. A further highly intense peak was detected at 1033 cm^−1^, which was associated with the stretching vibration of the aromatic alkyl C–O ether derived from the epoxydic ring opening of Polar Bear [61] as well.

The findings related to both the mechanical and chemical characterizations were further confirmed by microscopic investigations. Indeed, according to the acquired micrographs, when a low amount of rTP was added, the epoxy/rTP blend did not present a clear or well-distributed phase separation. This behavior can be explained by the mean of two different mechanisms: (1) the catalytic behavior of the hydroxyl groups towards the cross-linking reaction between part A and part B of the formulation; and (2) the cross-linking reaction between part A of the formulation, i.e., the epoxy prepolymer, and the rTP itself, which works as a curing agent. For rTP contents below or equal to 21%wt, the two aforementioned mechanisms are well balanced, thus achieving an optimized chemical reaction which ensures a high solubility of the rTP within the blend. Conversely, beyond the identified threshold value, the rTP starts to demix from the blend, causing the appearance of a clear and homogeneously distributed particulate morphology.

Figure 18 compares the cryofractured and etched surfaces for each formulation by increasing the rTP content at different magnifications. The increased percentage of the rTP added (27%wt) within the epoxy/thermoplastic blend leads to clear particulate morphologies with more and denser particles when compared to those of formulations with lower contents of rTP. For example, for the formulation containing 15%wt rTP, no clear second-phase domains can be distinguished, which is due either to their absence or their very small extensions.

The phase separation within the thermoplastic/epoxy blend containing 27%wt of rTP, which led to the worst thermo-mechanical properties found, is optically visible (Figure 19). Here, an increasing quantity of the rTP content added within the formulation determined the color change for the cured specimens, which turned from clear to pale brown. Two phases are evident in Figure 19c. This phenomenon is typical of a reaction-induced phase separation which occurs via the nucleus growth (NG) mechanism [62]. An analogous result was found by Yu et al. [63], who studied the effects of molecular weight on phase separation of PEI-modified TGDDM/DDS blends and discovered that different-molecular-weight PEI-modified systems may not have the same phase separation process. In the same way, the increase in the molecular weight of the rTP added within the formulation, i.e., contents up to 27%wt, triggered a phase separation phenomenon. Thus, in line with the NG mechanism, small thermoplastic particles aggregate into bigger ones during the curing process, and, in the end, a binary system showing island structures is obtained (see Figure 19c). Moreover, no significant phase separation phenomena were found for the blends containing an rTP content lower than 27%wt (see Figure 19a,b), which justify the enhanced thermo-mechanical properties found.

### 3.3. Optimization Process

The obtained results for the optimization process carried out by using the approach of desirability functions are shown in Figure 20. From these, it can be seen that the best-performing formulation in terms of its thermo-mechanical properties was the one corresponding to the configuration {A: 21%wt; B: C2}, since the value obtained for the desirability function corresponding to this configuration had the highest value, 0.8894, which is the nearest to the target value, 1.

## 4. Conclusions

A CE/C2C approach was used as a strategy to reuse recycled thermoplastic (rTP) derived from the epoxy matrix to be recycled into the virgin epoxy matrix itself. This approach had two advantage. First, we could devise a clear plan to define how to recycle and reuse the cured resin at its EoL, thus ensuring that once it enters the production cycle, it can be infinitely reused rather than just being recycled. Second, the chosen strategy permitted us to reduce the use of petroleum-based raw materials for the synthesis of the epoxy matrix, since variable contents (ranging between 15%wt and 27%wt) of recycled material were added to the virgin epoxy formulation.

Each of the investigated epoxy formulations containing a variable rTP content was thermo-mechanically tested to explore how the rTP’s incorporation affected the thermo-mechanical properties of the initial epoxy resin formulation. By analyzing each replicated general factorial design, it was possible to define how the rTP content and the implemented curing cycle affected each investigated thermal (T*_g_*) and mechanical property as determined by flexural tests (i.e., tests of the flexural strength, flexural modulus, and deformation at break). From these results, the following conclusions were found:homogeneous epoxy/thermoplastic blends with good thermo-mechanical properties were obtained for an rTP content not exceeding a threshold value of 27%wt;by implementing an additional post-curing process at 100 °C for 3 h, it was possible to enhance the thermo-mechanical properties;the best-performing epoxy system in terms of thermo-mechanical properties, as determined by an optimization process relying on the approach of desirability functions, was the one containing 21%wt rTP which was cured exploiting the C2 curing cycles.

Furthermore, it was highlighted that an increase in the rTP content within the virgin epoxy system caused a reduction in the T*_g_*, which still remained at an acceptable value (about 80 °C) for many applications when the additional post-curing step was used. For example, the commercial HexPly^®^ M79/M79-LT thermosetting system, which requires short cure cycles at a low temperature (about 100 °C) and is characterized by a T*_g_* equal to 95 °C, was designed for prepreg applications and to manufacture large industrial components. However, it is neither a bio-based nor a recyclable epoxy system. Conversely, the epoxy system identified as the best-performing one in terms of its thermo-mechanical properties corresponded to the configuration {A: 21%wt; B: C2}. As proposed in this experimental work, this could represent a novel thermoset system able to overcome the environmental issues related to thermosets’ typical non-recyclability and the use of petroleum-based raw materials for their synthesis.

Our recommendation for future work is an evaluation of rTP’s durability in recycling cycles, in terms of its chemical and thermal properties (i.e., its molecular weight, glass transition temperature, and so on), using the optimized epoxy/thermoplastic blends developed in this experimental work.

## Figures and Tables

**Figure 1 polymers-15-02809-f001:**
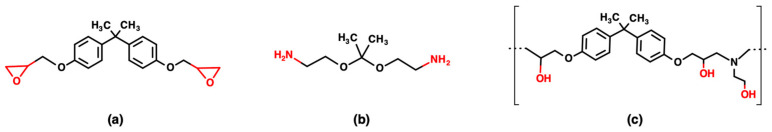
Chemical structures of (**a**) Polar Bear (part A), (**b**) Recyclamine^TM^ R*101 (**b**), and (**c**) recycled thermoplastic deriving from the chemical recycling process of the epoxy resin matrix (Polar Bear and Recyclamine^TM^ R*101) [14]. The reactive groups of each component are highlighted in red.

**Figure 2 polymers-15-02809-f002:**
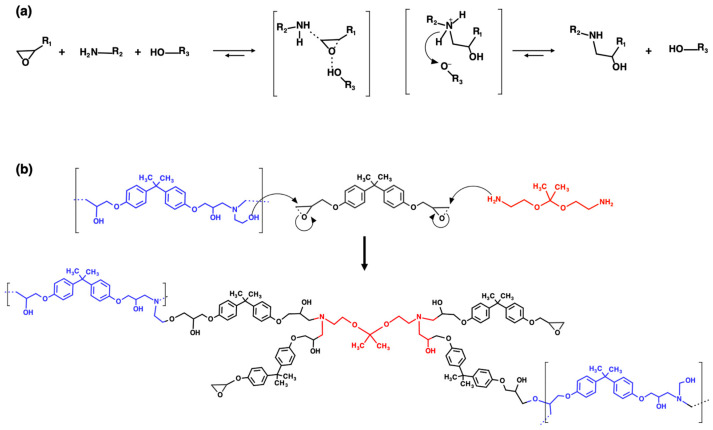
Main accepted mechanism of the reaction between epoxies and amines in presence of hydroxylic groups (**a**); reaction mechanism between rTP (blue structure), Polar Bear (black structure) and Recyclamine^TM^ R*101 (red structure) (**b**) [14].

**Figure 3 polymers-15-02809-f003:**
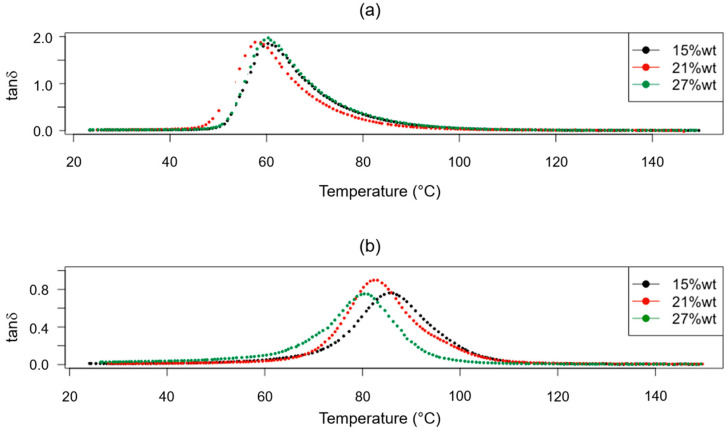
Obtained tanδ versus temperature curves for the investigated cross-linked thermoset systems exploiting different curing cycles: C1 (**a**) and C2 (**b**), and containing different rTP contents: 15%wt (black), 21%wt (red), and 27%wt (green).

**Figure 4 polymers-15-02809-f004:**
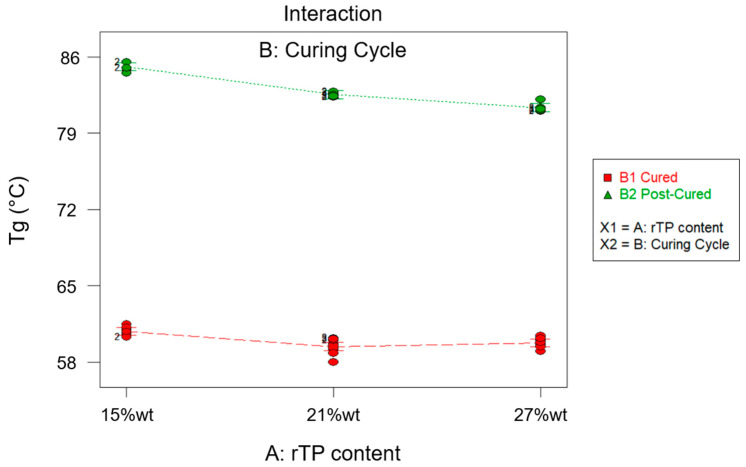
Effects diagram for glass transition temperature (T*_g_*).

**Figure 5 polymers-15-02809-f005:**
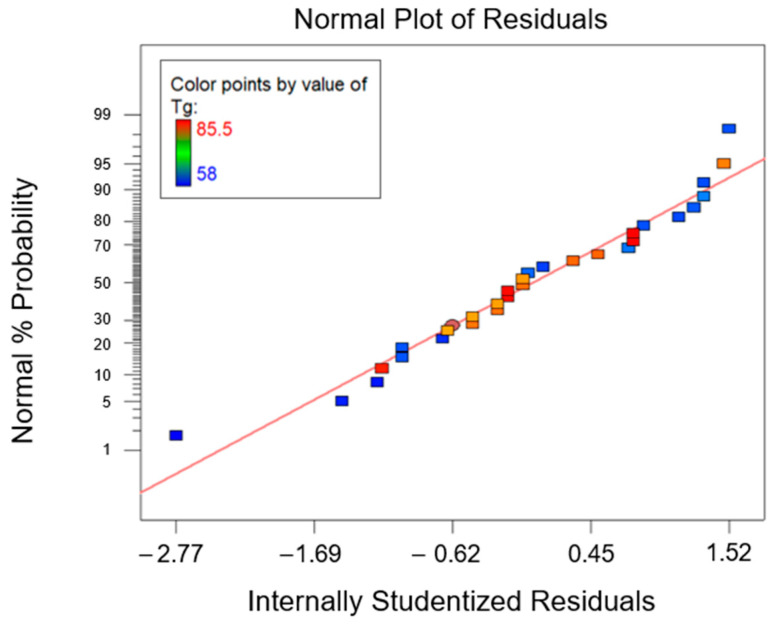
Normal probability plot for glass transition temperature (T*_g_*).

**Figure 6 polymers-15-02809-f006:**
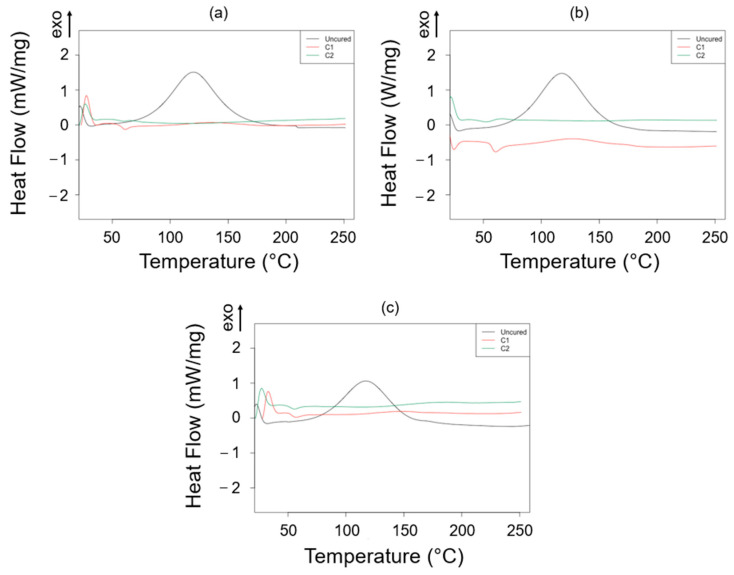
DSC thermograms acquired for the investigated thermoset systems containing different rTP contents: 15%wt (**a**), 21%wt (**b**), and 27%wt (**c**) and by considering three different curing states, i.e., uncured (black), cured (red), and post-cured (green).

**Figure 7 polymers-15-02809-f007:**
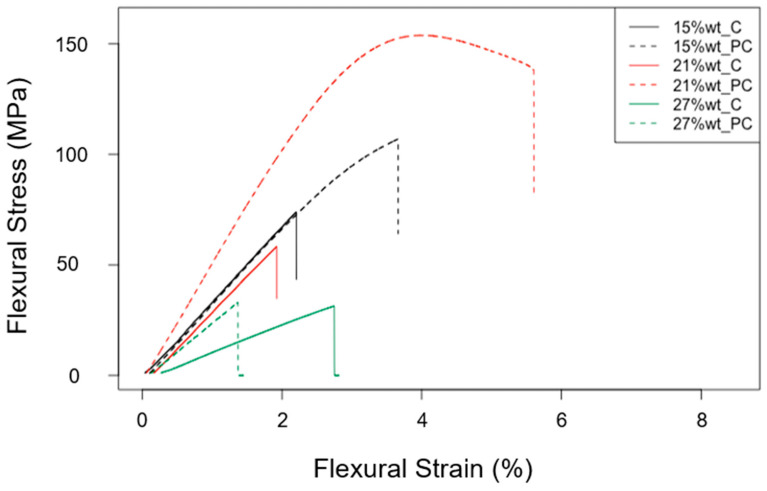
Representative flexural stress versus flexural strain curves determined for each epoxy resin formulation investigated (one sample per type).

**Figure 8 polymers-15-02809-f008:**
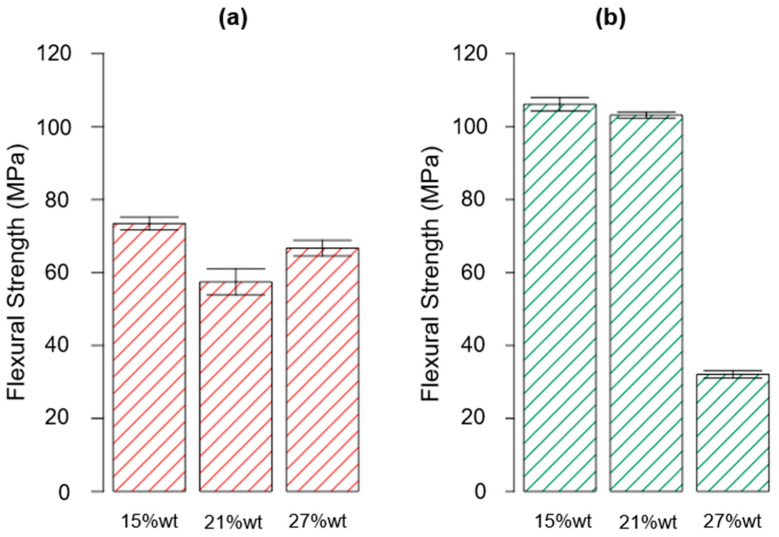
Bar plot showing the flexural strength for each investigated scenario, i.e., rTP content varying between 15, 21, and 27%wt and exploiting two different curing cycles: C1 (**a**) and C2 (**b**). Bar errors represent the standard deviation of each measure.

**Figure 9 polymers-15-02809-f009:**
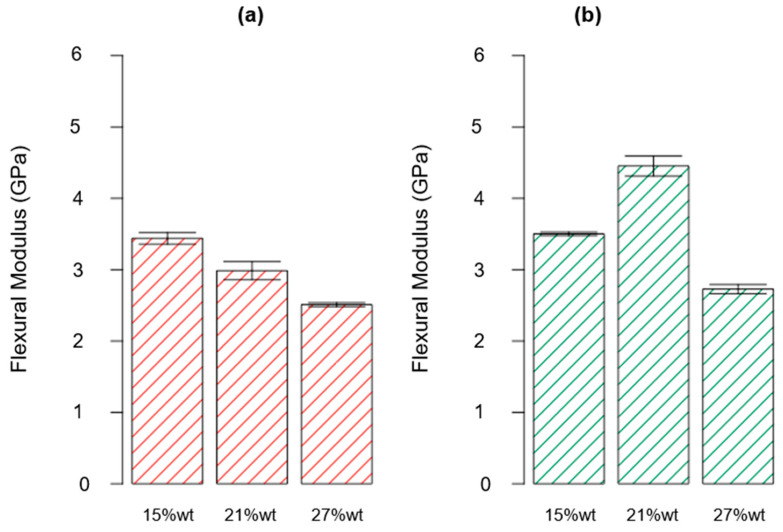
Bar plot showing the flexural modulus for each investigated scenario, i.e., rTP content varying between 15, 21, and 27%wt and exploiting two different curing cycles: C1 (**a**) and C2 (**b**). Bar errors represent the standard deviation of each measure.

**Figure 10 polymers-15-02809-f010:**
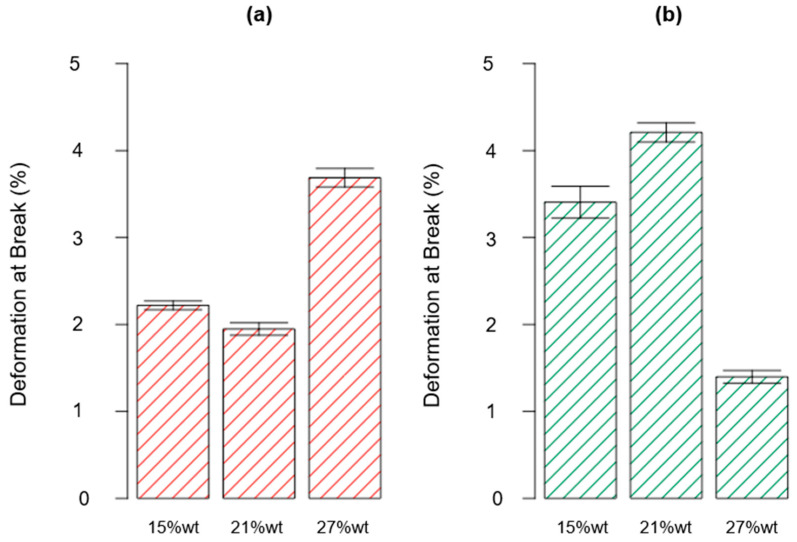
Bar plot showing the deformation at break for each investigated scenario, i.e., rTP content varying between 15, 21, and 27%wt and exploiting two different curing cycles: C1 (**a**) and C2 (**b**). Bar errors represent the standard deviation of each measure.

**Figure 11 polymers-15-02809-f011:**
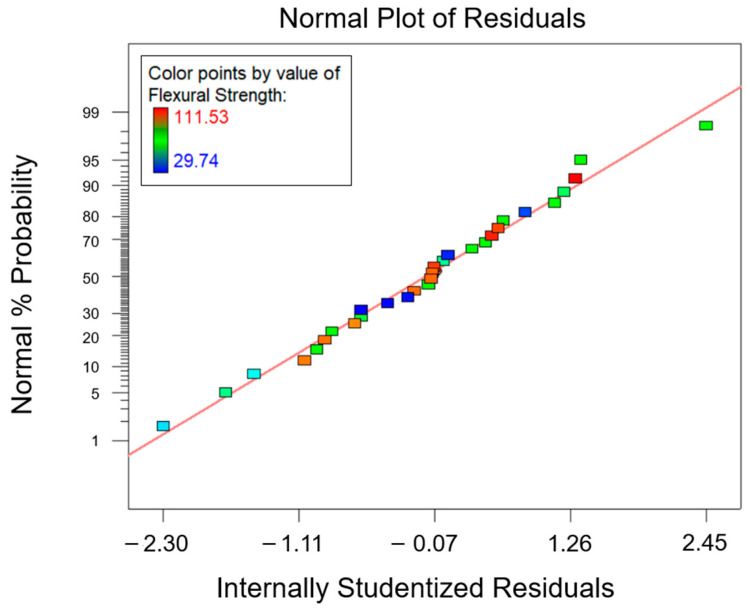
Normal probability plot for the flexural strength.

**Figure 12 polymers-15-02809-f012:**
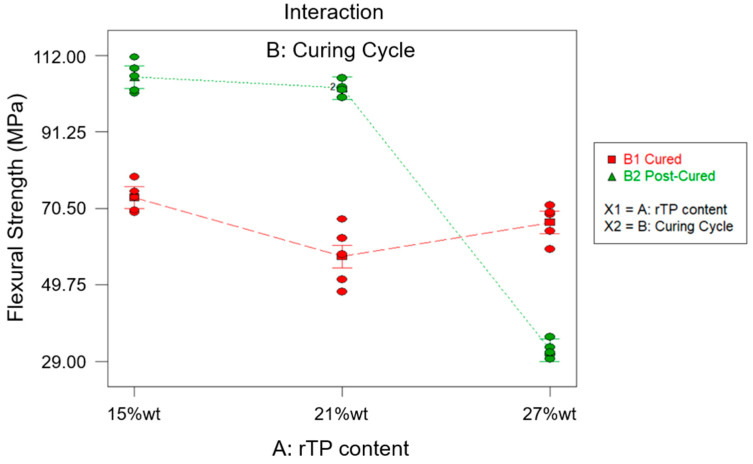
Effects diagram for flexural strength.

**Figure 13 polymers-15-02809-f013:**
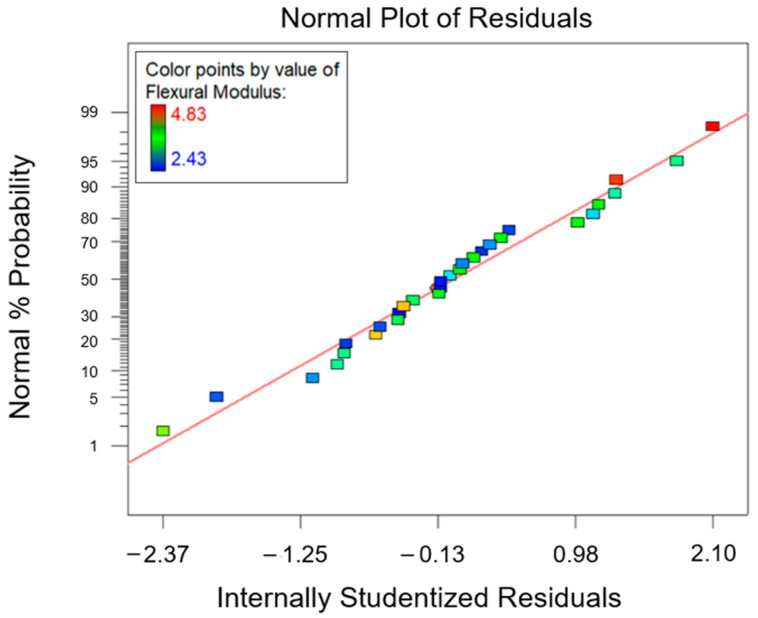
Normal probability plot for the flexural modulus.

**Figure 14 polymers-15-02809-f014:**
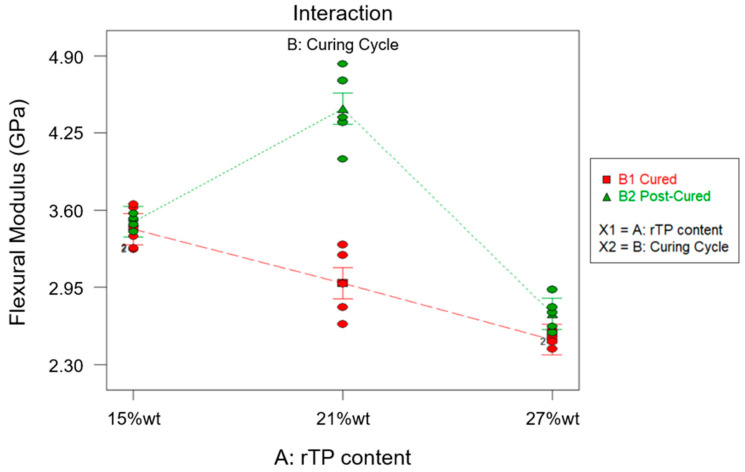
Effects diagram for flexural modulus.

**Figure 15 polymers-15-02809-f015:**
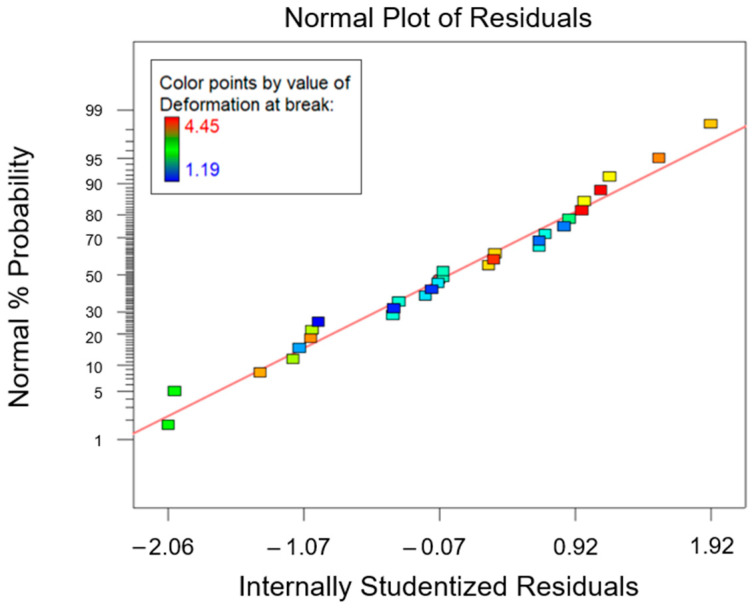
Normal probability plot for the elongation at break.

**Figure 16 polymers-15-02809-f016:**
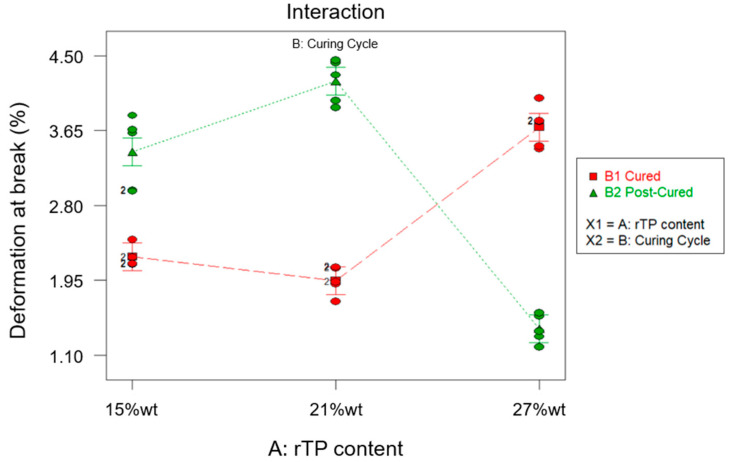
Effects diagram for the deformation at break.

**Figure 17 polymers-15-02809-f017:**
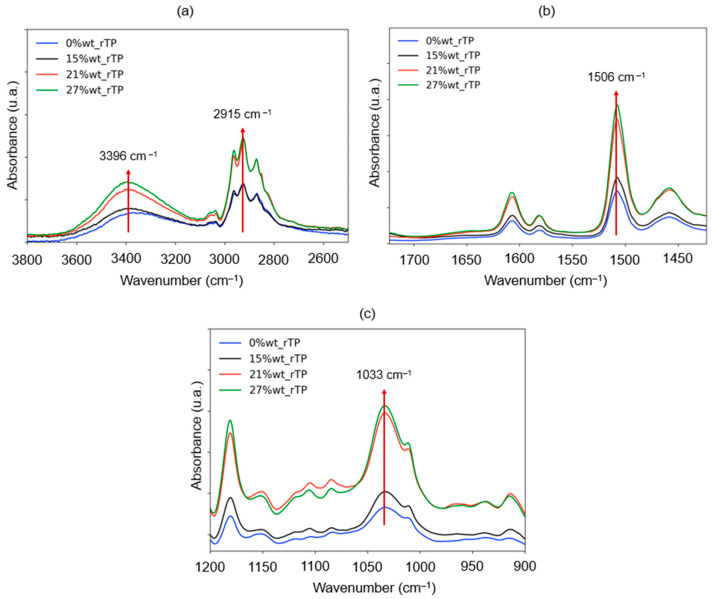
Evolution of the infrared spectra of cured formulations in three different ranges, i.e., 3800–2600 cm^−1^ (**a**), 1700–1450 cm^−1^ (**b**), and 1200–900 cm^−1^ (**c**), by varying the rTP content: 0%wt (blue), 15%wt (black), 21%wt (red), and 27%wt (green).

**Figure 18 polymers-15-02809-f018:**
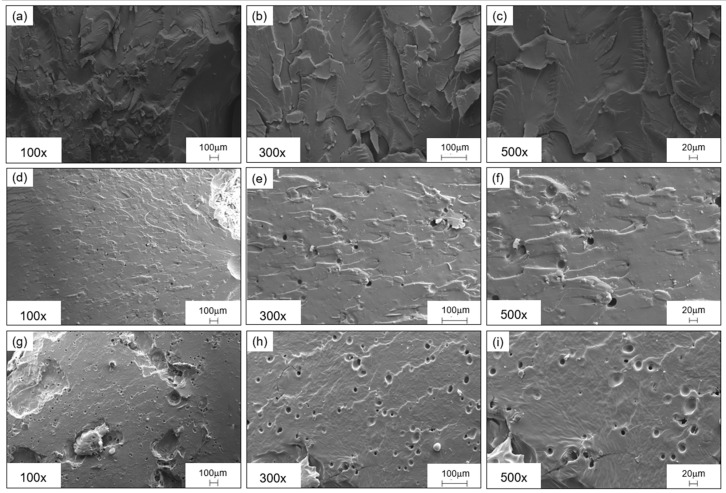
SEM micrographs of etched cryofractured surfaces for the epoxy/thermoplastic blends containing 15%wt rTP at a magnification of 100× (**a**), 300× (**b**), and 500× (**c**); 21%wt of rTP at a magnification of 100× (**d**), 300× (**e**), and 500× (**f**); 27%wt of rTP at a magnification of 100× (**g**), 300× (**h**), and 500× (**i**).

**Figure 19 polymers-15-02809-f019:**
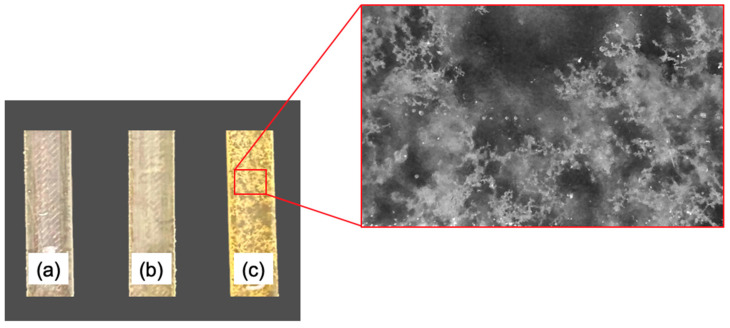
Cured specimens containing different contents of rTP: 15%wt (**a**), 21%wt (**b**), and 27%wt (**c**). In the red inset is reported micro-photograph of the cured specimen containing a 27%wt content of rTP, which was acquired by means of a digital microscope.

**Figure 20 polymers-15-02809-f020:**
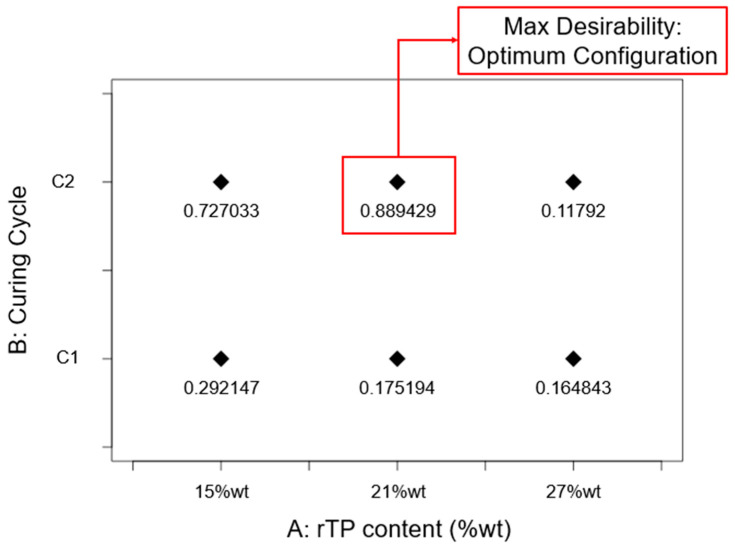
Desirability function value obtained for each investigated scenario. The red box identifies the optimum configuration in terms of thermo-mechanical properties among all the considered thermoset-modified formulations.

**Table 1 polymers-15-02809-t001:** Experimental plans. Factors, levels, and responses investigated.

Response	Factor	Symbol	Type	Unit	Level (−1)	Level (0)	Level (+1)
*T_g_*	rTP content	A	Categorical	(%wt)	15	21	27
	Curing cycle	B	Categorical	(-)	C1	-	C2
Flexural strength	rTP content	A	Categorical	(%wt)	15	21	27
	Curing cycle	B	Categorical	(-)	C1	-	C2
Flexural modulus	rTP content	A	Categorical	(%wt)	15	21	27
	Curing cycle	B	Categorical	(-)	C1	-	C2
Deformation at break	rTP content	A	Categorical	(%wt)	15	21	27
	Curing cycle	B	Categorical	(-)	C1	-	C2

**Table 2 polymers-15-02809-t002:** Set constraints for the optimization process based on the desirability functions.

Parameter	Goal	Lower Limit	Upper Limit	Lower Weight	Upper Weight	Importance
rTP content	In range	15%wt	27%wt	1	1	3
Curing cycle	In range	C1	C2	1	1	3
T*_g_*	Maximize	Lowest value collected	Highest value collected	1	1	3
Flexural strength	Maximize	Lowest value collected	Highest value collected	1	1	3
Flexural modulus	Maximize	Lowest value collected	Highest value collected	1	1	3
Deformation at break	Maximize	Lowest value collected	Highest value collected	1	1	3

**Table 3 polymers-15-02809-t003:** T*_g_* values collected from DMA analysis for each investigated formulation by varying the rTP content percentage added to the formulation and the curing cycle used.

Sample ID	Thermoset System	Curing Cycle	T*_g_* (°C)
P_A_rTP15_C	Polar + R*101 + 15%wt rTP	C1	60.80 ± 0.56
P_A_rTP15_PC	Polar + R*101 + 15%wt rTP	C2	85.10 ± 0.42
P_A_rTP21_C	Polar + R*101 + 21%wt rTP	C1	59.42 ± 0.98
P_A_rTP21_PC	Polar + R*101 + 21%wt rTP	C2	82.54 ± 0.21
P_A_rTP27_C	Polar + R*101 + 27%wt rTP	C1	59.76 ± 0.57
P_A_rTP27_PC	Polar + R*101 + 27%wt rTP	C2	81.34 ± 0.44

**Table 4 polymers-15-02809-t004:** ANOVA table for DMA test (investigated response: T*_g_*).

Source	Sum of Squares	df	Mean Square	F Value	*p*-Value	
**Model**	4007.42	5	801.48	2431.19	<0.0001	*significant*
**A–rTP content**	33.05	2	16.52	50.12	<0.0001	
**B–Curing Cycle**	3965.20	1	3965.20	12,027.91	<0.0001	
**AB**	9.17	2	4.59	13.91	<0.0001	
**Pure Error**	7.91	24	0.33			
**Cor Total**	4015.33	29				
**Std. Dev.**	0.57		**R-Squared**	0.9980		
**Mean**	71.50		**Adj R-Squared**	0.9976		

**Table 5 polymers-15-02809-t005:** Data obtained from the DSC analysis of the uncured blends by varying the rTP content.

Sample ID	Thermoset System	Curing State	Onset (°C)	Endset (°C)	Peak (°C)	Heat (J/g)
P_A_rTP15	Polar + R*101 + 15%wt rTP	Uncured	82.26	157.84	120.19	226.83
P_A_rTP21	Polar + R*101 + 21%wt rTP	Uncured	79.38	156.19	117.78	245.21
P_A_rTP27	Polar + R*101 + 27%wt rTP	Uncured	77.94	153.17	116.91	179.38

**Table 6 polymers-15-02809-t006:** Mechanical properties found from flexural tests for each investigated formulation by varying the percentage of rTP content added to the formulation and the curing cycle used (C1: at 25 °C for 24 h; C2: at 25 °C for 24 h + 100 °C for 3 h).

ID	Thermoset System	Flexural Strength (MPa)	Flexural Modulus (GPa)	Deformation at Break (%)
P_A_rTP15_C	Polar + R*101 + 15%wt rTP	73.44 ± 3.93	3.44 ± 0.18	2.22 ± 0.11
P_A_rTP15_PC	Polar + R*101 + 15%wt rTP	106.11 ± 4.13	3.50 ± 0.06	3.41 ± 0.41
P_A_rTP21_C	Polar + R*101 + 21%wt rTP	57.44 ± 8.04	2.99 ± 0.28	1.95 ± 0.16
P_A_rTP21_PC	Polar + R*101 + 21%wt rTP	103.16 ± 1.86	4.45 ± 0.31	4.21 ± 0.25
P_A_rTP27_C	Polar + R*101 + 27%wt rTP	66.68 ± 4.82	2.51 ± 0.06	3.69 ± 0.24
P_A_rTP27_PC	Polar + R*101 + 27%wt rTP	31.09 ± 2.32	2.73 ± 0.14	1.40 ± 0.16

**Table 7 polymers-15-02809-t007:** ANOVA table for flexural test (investigated response: flexural strength).

Source	Sum of Squares	df	Mean Square	F Value	*p*-Value	
**Model**	19,808.09	5	3961.62	184.02	<0.0001	*significant*
**A–rTP content**	8921.93	2	4460.97	207.22	<0.0001	
**B–Curing Cycle**	1598.85	1	1598.85	74.27	<0.0001	
**AB**	9287.31	2	4643.66	215.70	<0.0001	
**Pure Error**	516.67	24	21.52			
**Cor Total**	20,324.76	29				
**Std. Dev.**	4.64		**R-Squared**	0.9746		
**Mean**	73.16		**Adj R-Squared**	0.9693		

**Table 8 polymers-15-02809-t008:** ANOVA table for flexural test (investigated response: flexural modulus).

Source	Sum of Squares	df	Mean Square	F Value	*p*-Value	
**Model**	12.18	5	2.44	60.92	<0.0001	*significant*
**A—rTP content**	6.67	2	3.33	83.34	<0.0001	
**B—Curing Cycle**	2.55	1	2.55	63.66	<0.0001	
**AB**	2.97	2	1.48	37.12	<0.0001	
**Pure Error**	0.96	24	0.04			
**Cor Total**	13.14	29				
**Std. Dev.**	0.20		**R-Squared**	0.9270		
**Mean**	3.27		**Adj R-Squared**	0.9117		

**Table 9 polymers-15-02809-t009:** ANOVA table for flexural test (investigated response: elongation at break).

Source	Sum of Squares	df	Mean Square	F Value	*p*-Value	
**Model**	30.86	5	6.17	105.67	<0.0001	*significant*
**A—rTP content**	1.4	2	0.72	12.30	0.0002	
**B—Curing Cycle**	1.12	1	1.12	19.14	0.0002	
**AB**	28.30	2	14.15	242.31	<0.0001	
**Pure Error**	1.40	24	0.06			
**Cor Total**	32.26	29				
**Std. Dev.**	0.24		**R-Squared**	0.9566		

## Data Availability

Data will be made available on request.

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
