# Peer review of "Chemical Recycling of Fully Recyclable Bio-Epoxy Matrices and Reuse Strategies: A Cradle-to-Cradle Approach"

_polymers, 2023, doi:10.3390/polym15132809_

Round 1
Reviewer 1 Report
The manuscript written by Saita et al. titled “Chemical Recycling of Fully-recyclable Bio-Epoxy Matrices and Re-use strategies: a Cradle-to-Cradle approach” provided a study of the reuse of epoxy-amine thermoset degradation fragments as a reactive crosslinker to the virgin epoxy-amine formulation. After going through the article, I would like to say that the article is well written and the figures are well presented.
After going through the manuscript, I want to suggest the following comments and clarify a few concerns which will further improve the article and make it suitable for Polymers.
1. Please provide the specifications of the epoxy and amine crosslinker like EEW and AHEW, etc.
2. If the recycled part can react with the epoxy resin, will it be appropriate to call it thermoplastic? Also, any evidence for the catalytic activity of the rTP during curing?
3. Although the authors concluded the possibility of reuse for infinite time, after each application only 21% rTP can be used in the virgin resin, so what about the remaining acid hydrolyzed rTP?
4. The authors should measure the acid durability of the cured resin as the presence of the acetal groups introduced to degrade the network might also prevent the epoxy resin from use in acidic industrial environment.
5. Study of the cured networks from BPA-based epoxy resin at various temperatures and time intervals was carried out by Webster et al. The author could mention them in the introduction. https://doi.org/10.1016/j.polymer.2021.124191; https://doi.org/10.1016/j.porgcoat.2023.107471.
Author Response
We thank the reviewer for his/her valuable comments. Please find attached the file with our responses.

Reviewer 2 Report
This manuscript is concerned with the usage of recycled epoxy resins as modifiers for new epoxy resins by a Cradle-to-Cradle approach. The authors investigated the effects of recycled epoxy resin contents on the curing behaviors, thermal and mechanical properties of resulting modified epoxy resins. This manuscript is well organized and easy to follow. I only have some minor comments, listed below.
1. The authors should clearly show that rTPs with different contents are dissolved or dispersed in the epoxy matrix.
2. What’s the effect of rTPs with different contents on the viscosity of the resulting epoxy systems, which is closely related to the processability of epoxy resins.
3. In the Results and Discussion section, the DSC results are suggested to be discussed in front of DMA.
4. Why does P_A_rTP21 show increased curing heat in Table 5?
5. The epoxy sample with 15% rTP shows the best flexural performance after post cure as shown in Table 6. This means that a lower content of rTP (<15%) is also needed to study, and thus the trend of the effect of rTP contents on the mechanical properties of resulting epoxy resins would be clear.
The language of this manuscript should be further polished.
Author Response

(The authors gave the same response as above.)

Reviewer 3 Report
In this contribution, Saitta et al. reported a strategy to reuse the thermoplastics derived from the epoxy matrix recycling into the virgin epoxy matrix itself, and the effect of epoxy formulation on the properties of recycled polymers were studied. Generally speaking, this is a relatively systematic work, but the following problems need to be solved.
1. This manuscript is rich in volume. It can be seen that the authors have done a lot of work. But I think some languages need further refinement. It is suggested that the author should further polish the language.
2. Some format issues should be noted. For example, page 5, line 210, omitted half brackets. The writing of Tg should be unified.
3. In the part of results and discussion, it is suggested to make some summary descriptions rather than simply pile up experimental phenomena.
4. The authors selected 15% - 27% rTP formulation. Could the author explain why this value was selected. In addition, the influence direction of rTP content on different properties of polymers was inconsistent. It is suggested to discuss the reasons behind it.
5. Figure 2. The hydroxyl group in red was from epoxy. Should it be marked in black?
6. The reuse of polymer after degradation is a hot field recently, and some relevant references are suggested to be cited. For example, Macromolecules, 2022, 55, 1726-1735. Nat. Sustain. 2023, DOI: 10.1038/s41893-023-01082-z.
There were too many long sentences in the manuscript, and some of them were a little wordy. It is recommended to polish the language.
Author Response

(The authors gave the same response as above.)

Reviewer 4 Report
In this Manuscript entitled “Chemical Recycling of Fully-recyclable Bio-Epoxy Matrices and Re-use strategies: A Cradle-to-Cradle approach”, the authors aimed to exploit a Cradle-to-Cradle approach, according to a complete circular economy model. They reused the recycled polymers by mixing them with the same bio-based epoxy formulations with varying percentages (15 wt.% to 27 wt.%). The mechanical properties of these formulations were evaluated by using dynamic-mechanical analysis, differential scanning calorimetry, and flexural testing. Finally, the authors used general factorial design as a statistical tool to find out the right percentage of recycled polymer with the most performing epoxy matrix formulation in terms of thermo-mechanical properties.
Overall, some issues are associated with this research article, which need to be addressed before possible publication.
Please find the attached annotated file to see my comments.
Lastly, I would like to say Polymers Journal publishes high-quality research articles related to the recycling of polymers. Based on my comments mentioned in the annotated file, the recommendation is Major Revision.

Sentence structuring should be improved.
Author Response
We thank the reviewer for his/her valuable comments. Please find attached the file with our responses to your comments on the pdf version of our original manuscript.
All the modification in the uploaded revised manuscript, according to your valuble comments and suggestions, are highlighted in yellow.

Round 2
Reviewer 2 Report
Since the authors have addressed all my concerns, I think the revised manuscript is ready for publication.
Reviewer 4 Report
Now, the article is in acceptable form.
Minor editing of English language required.